# Dopamine-Mediated Graphene Bridging Hexagonal Boron Nitride for Large-Scale Composite Films with Enhanced Thermal Conductivity and Electrical Insulation

**DOI:** 10.3390/nano13071210

**Published:** 2023-03-29

**Authors:** Shikun Li, Yutan Shen, Xiao Jia, Min Xu, Ruoyu Zong, Guohua Liu, Bin Liu, Xiulan Huai

**Affiliations:** 1Beijing Key Laboratory of Multiphase Flow and Heat Transfer for Low Grade Energy Utilization, North China Electric Power University, Beijing 102206, China; lishikun@iet.cn (S.L.); liuguohua126@126.com (G.L.); 2Institute of Engineering Thermophysics, Chinese Academy of Sciences, Beijing 100190, Chinaxumin@iet.cn (M.X.); zongruoyu@iet.cn (R.Z.); 3Nanjing Institute of Future Energy System, Nanjing 211135, China; 4SINOPEC Research Institute of Petroleum Processing Co., Ltd., Beijing 100083, China; shenyutan.ripp@sinopec.com

**Keywords:** anisotropic thermal conductivity, electrical insulation, hybrid filler, large-scale, dopamine

## Abstract

Heat accumulation generated from confined space poses a threat to the service reliability and lifetime of electronic devices. To quickly remove the excess heat from the hot spot, it is highly desirable to enhance the heat dissipation in a specific direction. Herein, we report a facile route to fabricate the large-scale composite film with enhanced thermal conductivity and electrical insulation. The well-stacked composite films were constructed by the assembly of polydopamine (PDA)-modified graphene nanosheets (GNS_PDA_) and hexagonal boron nitride (BN_PDA_), as well as bacterial cellulose (BC). The introduction of the PDA layer greatly improves the interface compatibility between hybrid fillers and BC matrix, and the presence of GNS_PDA_-bridging significantly increases the probability of effective contact with BN_PDA_ fillers, which is beneficial to form a denser and complete “BN-GNS-BN” heat conduction pathway and tight filler–matrix network, as supported by the Foygel model fitting and numerical simulation. The resulting BC/BN_PDA_/GNS_PDA_ film shows the thermal conductivity and tensile strength of 34.9 W·m^−1^·K^−1^ and 30.9 MPa, which separately increases to 161% and 155% relative to the BC/BN_PDA_ film. It was found that the low electrically conductive and high thermal conductive properties can be well balanced by tuning the mass ratio of GNS_PDA_ at 5 wt%, and the electrical conductivity caused by GNS_PDA_ can be effectively blocked by the BN_PDA_ filler network, giving the low electrical conductivity of 1.8 × 10^−10^ S·cm^−1^. Meanwhile, the BC/BN_PDA_/GNS_PDA_ composite films effectively transfer the heat and diminish the hot-spot temperature in cooling LED chip module application. Thus, the present study may pave the way to promoting the industrialization of scalable thermal management devices.

## 1. Introduction

More and more attention has been given to the fact that heat accumulation generated in a confined space poses a threat to the service reliability and lifetime of electronic components, especially for the integration and densification of electronic and electrical devices [1,2,3,4]. To quickly transfer the excess heat from electrical and electronic devices, it is required to enhance the heat dissipation in a specific direction [5,6,7,8]. At the same time, electrical insulation is a prerequisite to ensure the stable and secure operation of thermal management materials in the electric power field [9,10].

Hexagonal boron nitride (BN) possesses excellent dielectric breakdown strength (35 kV·mm^−1^) and high thermal conductivity (180 W·m^−1^·K^−1^) [11], which is an ideal filler for the fabrication of the highly thermal conductive and electrical insulation composite materials. For example, BN has been used for embedding into the polymer matrix (comprising natural rubber [12], epoxy resin [13], and cellulose nanofibers [14]) to fabricate functional composite films [15]. However, the existence of interfacial thermal resistance (*R*_I_) significantly affects the heat transfer efficiency along BN filler pathways, for example, the interfaces between BN fillers or between filler and matrix. Thus, many studies have pointed out that the *R*_I_ can be reduced by enhancing the interface compatibility through an interface modification strategy [16,17]. In fact, the improvement extent of thermal conductivity also has certain limitations only when reducing the interfacial thermal resistance. This is because the thermal contact resistance (*R*_c_) caused by the poor contact between fillers is another key parameter that determines the final thermal conductivity of composites. It is reported that the contact probability can be effectively increased by introducing another additional filler instead of using only one filler component [18,19]. For instance, the zero-dimensional silver nanoparticles and one-dimensional silicon carbide nanowires were selected to connect BN fillers [20,21]. Compared with the “point-to-point” contact type of zero/one-dimensional materials, two-dimensional materials provide the “face-to-face” contact sites between the thermal conductive fillers [22], which seem to be a very effective strategy for improving effective contact and building a bridge of fillers.

As the most potential two-dimensional material in the thermal management field, graphene nanosheets (GNS) have exceptional thermal conductivity (600–5000 W·m^−1^·K^−1^) and tensile strength (130 GPa) [23]. Nevertheless, this kind of ultra-high electrical conductivity (10^6^ S·m^−1^) makes it hard to directly apply it to the thermal management devices of electronics and the power industry which require good electrical insulation [24]. So, GNS can be used as a candidate for other, Appendix A of the main BN filler in order to exploit hybrid composites with excellent thermal conductivity and dielectric properties. However, it is difficult to uniformly disperse two heterogeneous fillers due to the poor interface compatibility, resulting in numerous agglomeration and serious phase separation problems. Moreover, even the most extensive methods of preparing composites, such as hot pressing, vacuum filtration, and freeze-drying, are still far from the industrialized production of scalable and highly arranged composites [25,26,27]. Therefore, it is highly desirable for the fabrication of large-scale and regularly oriented hybrid composite materials with excellent insulation and thermal conductivity through a convenient and facile approach.

Here, we put forward the dopamine-mediated graphene bridging of hexagonal boron nitride to fabricate the large-scale composite films with enhanced thermal conductivity and electrical insulation. As depicted in Figure 1, as a biodegradable and naturally recyclable material, dopamine (DA) could easily self-polymerize into polydopamine (PDA) and adhere to the surface of BN and GNS fillers. As another degradable biopolymer, bacterial cellulose (BC), with numerous hydroxyl groups was expected to connect PDA modified hybrid BN and GNS fillers via multiple noncovalent interactions. Finally, the large-scale BC/BN_PDA_/GNS_PDA_ composite films were prepared through a layer-by-layer (LBL) scraping method. The results demonstrate that the flat and highly directional microstructure of BC/BN_PDA_/G_PDA_ composite films were successfully built, and the high anisotropic thermal conductivity, enhanced tensile strength, and excellent electrical insulation performances can be effectively balanced by regulating the critical value of the mass ratio of GNS_PDA_ fillers. It was found that the main BN_PDA_ fillers can be bridged by flexible GNS_PDA_ filler to build a continuous and highly oriented thermally conductive pathway under the optimal mass ratio of GNS_PDA_ of 5 wt%; meanwhile, the electrical conductivity arising from GNS_PDA_ can also be effectively blocked by the electrical insulation of h-BN filler. Furthermore, the BC/BN_PDA_/GNS_PDA_ films show high efficiency in transferring the hot-spot heat from the LED chip, demonstrating great potential applications as thermal management electronic devices.

## 2. Experimental Section

### 2.1. Materials

BN (99%, diameter ~7 μm) was purchased from Macklin Biochemical Co., Ltd. (Shanghai, China), and GNS (97%, diameter ~7 μm) was purchased from Kanao Graphene Technology Co., Ltd. (Shenzhen, China). BC (0.73 wt%) was purchased from Qihong Technology Co., Ltd. (Guilin, China). Dopamine (98%), tris(hydroxymethyl)-aminomethane hydrochloride (Tris-HCl) and ethanol were purchased from Beijing Chemical Reagent Co., Ltd. (Beijing, China).

### 2.2. Preparation of Composite Films

A total of 3 g BN (or GNS) filler and 1.2 g dopamine were added into 500 mL Tris-HCl buffer solution (pH = 8.5). After stirring at room temperature for 24 h, the mixture was filtered and washed with ethanol three times, and the PDA modified BN and GNS fillers were represented as BN_PDA_ and GNS_PDA_, respectively. Then, the BN_PDA_ and GNS_PDA_ fillers were obtained after drying for 12 h at 60 °C, and the desired content ratio of BN_PDA_ and GNS_PDA_ fillers were added into BC solution (2 mg/mL) and stirred for 3 min. Subsequently, the mixture was cast onto nylon film and heated at 100 °C for 5 min. Finally, the large-scale films were obtained using the layer-by-layer (LBL) scraping method. The composite films composed of BC and hybrid fillers were represented as BC/BN_PDA_/GNS_PDA_-*X*, and *X* is the mass content of GNS_PDA_ in the hybrid fillers (*X* = 0, 1, 3, 5, 7, and 10).

### 2.3. Characterization

The morphologies and energy dispersive X-ray spectrometry (EDS) images of filler, matrix and composite films were characterized by emission scanning electron microscopy (SEM, Hitachi, S-4800, Shimadzu, Tokyo, Japan). The Raman spectroscopy of DA, BN, BN_PDA_, GNS, and GNS_PDA_ fillers was performed on a Raman spectroscope (inVia Reflex, Renishaw, London, UK) with a 532 nm laser excitation wavelength. The thermal degradations of BN, BN_PDA_, GNS, and GNS_PDA_ fillers were tested by thermogravimetry (TGA, INNUO TGA-1000, Yingnuo, Shanghai, China) under a nitrogen atmosphere (10 °C·min^−1^, 50–800 °C). X-ray photoelectron spectra (XPS) recorded the chemical composition of BN, BN_PDA_, GNS, and GNS_PDA_ fillers on an ESCA instrument (Physical Electronics, MA, USA). Isothermal titration calorimetry (ITC) experiments were performed with a Microcal VP-ITC apparatus at 298.15 K. An X-ray diffractometer (XRD, Rigaku SmartLab 9 kW, Japan) analyzed the orientation of hybrid fillers with a diffraction angle in the range of 2–90°. The mechanical properties of the composite films were tested using a tensile testing machine (AI-7000-LAU10, Gotech Testing Machines Co., Ltd., Dongguan, China). Dielectric properties and the AC conductivity were acquired by the Tong Hui TH26077 Precision LCR Meter in the frequency range of 1000 Hz to 10 MHz. Thermal images were recorded by an infrared camera (Ti400, Fluke, Everett, WA, USA). The numerical simulation software (ANSYS, 16.0) was employed to simulate the heat transfer characteristics of composite films. The anisotropic thermal conductivities of composite films were calculated by the equation: *k* = *α* × *ρ* × *C*, where *α*, *ρ*, and *C*, respectively, correspond to the thermal diffusivity, density, and specific heat capacity of composite films, which is measured by the laser flashing method (LFA 467, NanoFlash, Netzch, Germany). C was determined through the following equation: *C* = *C*_BN_ × *ω* + *C*_BC_ × (1 − *ω*), where *ω* is the filler content and *C*_BN_ and *C*_BC_ were separately measured by a differential scanning calorimeter (NETZSCH DSC214, Germany).

## 3. Results and Discussion

### 3.1. Characterizations of PDA Modified BN, GNS, and Composite Films

The microstructures and size distributions of BN and GNS fillers are shown in Appendix A. It can be observed that the BN and GNS fillers have a similar filler size (~7 μm), and the GNS filler shows obvious flexibility compared with the brittle BN filler. Meanwhile, Appendix A displays the SEM images of BC nanofibers, and the inset shows an optical photo of BC solution. It can be seen that the BC nanofibers exhibit a particular ultrafine network structure. The Raman spectra of the DA exhibits the obvious characteristic peak of hydroxyl and amino groups appear separately at 1286 and 1616 cm^−1^ (Appendix A) [28]. Figure 1a displays Raman spectra of the original BN, and GNS fillers and modified BN_PDA_ and GNS_PDA_ fillers. The characteristic peak of BN filler appears at 1364 cm^−1^ [10]. After the PDA modification, a new absorption peak appears near 1557 cm^−1^, which is attributed to the hydroxyl groups arising from PDA layer on the surface of the BN filler [22]. The Raman shifts at 1263 (weak D-band) and 1575 cm^−1^ (strong G-band) are the characteristic peaks of GNS [24]. After the introduction of PDA, the D-band peak shifts to 1266 cm^−1^ while the G-band peak decreases to 1564 cm^−1^, which is attributed to the deformation of the catechol groups of the PDA [6]. Figure 1b shows the TGA curves of BN and GNS before and after PDA modification, and BN has no thermal weight loss in the temperature range of 50 to 800 °C, whereas the BN_PDA_ shows thermal degradation between 300 and 650 °C and weight loss reaches 1.89 wt% at 800 °C [5]. GNS and GNS_PDA_ both exhibit obvious thermal degradation, but GNS_PDA_ has a greater weight loss of 3.68 wt% due to the PDA-coated layer. Figure 1c presents the XPS spectra of PDA modified fillers. Compared with the untreated BN filler, the peak intensities of the O and C elements increase but the B and N elements are diminished [5]. For GNS_PDA_, the N element appears relative to the untreated GNS. In addition, ITC was employed to investigate the thermodynamic behavior of the interactions between BN (or GNS) and PDA (Figure 1d) [5,29]. When the PDA solution was titrated into the BN (or GNS) solution, the observed enthalpy changes (Δ*H*_obs_) are negative, indicating that the combination of PDA and BN (or GNS) is accompanied by exothermic process. The Δ*H*_obs_ value gradually decreases to zero plateau with the addition of PDA solution, which manifests the saturation of the noncovalent binding between PDA and BN or (GNS). The binding constants (*K*_d_) of PDA with BN or GNS can be derived by fitting the ITC curves, and the resulting *K*_d_ values give 1.14 × 10^3^ and 3.53 × 10^3^ M^−1^, respectively, which demonstrates that PDA and BN (or GNS) are effectively combined, and these two fillers have a comparable binding ability with PDA. In brief, this evidence proves that BN and GNS fillers are successfully modified by PDA.

The cross-section of four films, pure BC (Appendix A), BC/BN_PDA_ (the mass ratio of BN fillers is 70 wt%), BC/BN_PDA_/GNS_PDA_-5 (the total mass ratio of hybrid fillers is 70 wt% and the mass ratio of GNS_PDA_ is 5 wt%), and BC/BN_PDA_/GNS_PDA_-10 (the total mass ratio of hybrid fillers is 70 wt% and the mass ratio of GNS_PDA_ is 10 wt%) were observed from SEM. Figure 2a presents the cross-sectional morphology of the BC/BN_PDA_ film, which shows the partially oriented stacking of BN_PDA_ fillers (as revealed by the red arrows) [5]. After the introduction of 5 wt% GNS_PDA_, the cross-section of BC/BN_PDA_/GNS_PDA_-5 film exhibits more regular arrangement and forms longer networks of hybrid fillers relative to BC/BN_PDA_ film, as shown in Figure 2b. This may be because GNS_PDA_ plays a key role in face-to-face contact with the BN_PDA_ filler, making the formation a continuous network structure inside the BC/BN_PDA_/GNS_PDA_-5 film. Simultaneously, the energy dispersive spectrometer (EDS) mapping of the BC/BN_PDA_/GNS_PDA_-5 film (Figure 2d) clearly shows the lamellar distributions of B, N, and C elements, which confirms the well-stacked layered structure of the hybrid fillers. Moreover, when the mass ratio of GNS_PDA_ increases to 10 wt%, the uniform and close networks of BC/BN_PDA_/GNS_PDA_-10 film appear in Figure 2c. Additionally, this result indicates that the probability of effective contact between hybrid fillers significantly increases, which is beneficial for forming a denser and complete “BN-GNS-BN” pathway.

To further to evaluate the orientations of BN fillers, XRD measurements on BC/BN_PDA_ and BC/BN_PDA_/GNS_PDA_-5 were performed. As shown in Figure 2e, two characteristic peaks arising from BC nanofiber appear at 15° and 23°. Moreover, the characteristic peaks at 27° and 42° arise from the (002) and (100) planes of the BN fillers, and the intensities of these two peaks are expressed as *I*_002_ and *I*_100_. It is reported that the increased values of *I*_002_/*I*_100_ means the improved in-plane orientation of BN fillers for the composite film [20]. Take the example of the BC/BN_PDA_/GNS_PDA_-5 film: the *I*_002_/*I*_100_ value of BC/BN_PDA_/GNS_PDA_-5 film gives 132, which is obviously higher than that of BC/BN_PDA_ films (89) [5]. Thus, by the aid of the flexible GNS_PDA_, the orientation of BN fillers along the in-plane direction is greatly enhanced. This result is consistent with the observed morphology features, which shows that the BC/BN_PDA_/GNS_PDA_ film exhibits increased in-plane interconnected structures relative to BC/BN_PDA_ films. In general, the distribution of GNS_PDA_ in the hybrid fillers is closely related to the formation of thermal contact channels in composite films, which directly determines the heat transfer efficiency. As illustrated in Figure 2f–h, the BC/BN_PDA_ film shows randomly oriented BN networks. After the introduction of GNS_PDA_ filler, the BN_PDA_ fillers can be connected by GNS_PDA_ fillers to form the more continuous and longer “BN-GNS-BN” pathways (Figure 2g,h), which are beneficial for filling the interfacial gap between BN and BC and improving the insufficient contact between the brittle BN fillers. Therefore, by introducing a small amount of GNS_PDA_ filler, the interface compatibility between hybrid fillers and BC matrix is improved, and more importantly, the possibility of effective contact between hybrid fillers is also enhanced, which is expected to diminish the heat transfer resistance in the interfacial thermal resistance (*R*_I_) and thermal contact resistance (*R*_c_).

### 3.2. Thermal Conductivity of Composite Films

The continuous and regularly oriented heat conduction pathway is the core of heat transfer process. The anisotropic thermal conductivities of different composite films at varied hybrid fillers contents were measured through the LFA method (details described in Appendix A). In our previous work, the thermal conductivity of BC/BN_PDA_ films obtained by vacuum filtration are 21.6 W·m^−1^·K^−1^ [5], and these values are close to the results of BC/BN_PDA_ films prepared using the LBL scraping method (21.7 W·m^−1^·K^−1^). Figure 3a shows that the BC/BN_PDA_/GNS_PDA_-5 composite films always exhibit higher in-plane thermal conductivities than BC/BN_PDA_ films at varied filler contents. In particular, the in-plane thermal conductivity of the BC/BN_PDA_/GNS_PDA_-5 composite film reaches 34.9 W·m^−1^·K^−1^ at the hybrid filler mass ratio of 70 wt%, which increases to 161% relative to the BC/BN_PDA_ film (21.7 W·m^−1^·K^−1^) [5]. By contrast, Figure 3b displays the through-plane thermal conductivities of the BC/BN_PDA_/GNS_PDA_-5 composite films, which are of lower values than those of BC/BN_PDA_ films at different filler contents. This may be because the introduction of a small amount of GNS_PDA_ (5 wt%) plays an important role in bridging the main BN filler (95 wt%), which greatly increases the contact of BN fillers along the in-plane direction, as observed from the continuous and well-stacked BN structure in Figure 2b, which is conducive to enhancing the anisotropic heat transfer efficiency of BC/BN_PDA_/GNS_PDA_-5 composite films. The effect of GNS_PDA_ contents on the in-plane thermal conductivities of the BC/BN_PDA_/GNS_PDA_ films (the total content of hybrid fillers is fixed at 70 wt%) is also considered. Since the high content of GNS will lead to poor electrical insulation performance [6], the influence of GNS_PDA_ contents in the range of 0~10 wt% on the in-plane thermal conductivity is investigated in Figure 3c. It can be clearly observed that the in-plane thermal conductivity of the composite film has a significant increase when the mass ratio of GNS_PDA_ reaches 5 wt%, but the improvement extent of in-plane thermal conductivities is limited by further increasing the mass ratio of GNS_PDA_. Therefore, the synergistic effect between GNS_PDA_ and BN_PDA_ may exist in improving heat transfer efficiency at the critical GNS_PDA_ content of 5 wt%.

It Is essential to elucidate the effect of a small mass ratio of GNS_PDA_ (5 wt%) on the thermal conductivity of the composite films, and thus, the effective medium theory (EMT) [25] and Foygel model [10] were used to investigate their influence on interfacial thermal resistance (*R*_I_) and thermal contact resistance (*R*_c_) of inner composite films. According to the in-plane thermal conductivity of the composite films, the EMT model is employed to calculate the *R*_I_ values between filler and BC matrix before and after the introduction of GNS_PDA_, and the corresponding equations are shown below:(1)$$k=k_{m}\frac{3+f\left(\beta_{⊥}+\beta_{∥}\right)}{3-f\beta_{⊥}}$$
(2)$$\beta_{⊥}=\frac{2\left(d\left(k_{f}-k_{p}\right)-2R_{I}k_{f}k_{p}\right)}{d\left(k_{f}+k_{p}\right)+2R_{I}k_{f}k_{p}}$$
(3)$$\beta_{∥}=\frac{L\left(k_{f}-k_{p}\right)-2R_{I}k_{f}k_{p}}{Lk_{p}+2R_{I}k_{f}k_{p}}$$
where *k*_p_ (3.05 W·m^−1^·K^−1^, Appendix A), *k*_f_ (180 W·m^−1^·K^−1^), and *k* are the in-plane thermal conductivities of BC matrix, BN fillers, and composite films, respectively; *f* is the volume fraction of filler; *d* (0.5 μm) and *L* (7.0 μm) are the thickness and lateral size of BN (Appendix A); and *R*_I_ is obtained by using the EMT equations (Appendix A). The resulting *R*_I_ of BC and BN_PDA_/GNS_PDA_-5 gives 1.83 × 10^−8^ m^2^·K·W^−1^, which is evidently lower than the BC and BN_PDA_ (3.06 × 10^−8^ m^2^·K·W^−1^). These results demonstrate that a small amount of GNS_PDA_ (5 wt%) eliminates interfacial gaps, thereby reducing the interfacial thermal resistance between the filler and the BC matrix.

The Foygel model was further employed to clarify the introduction of flexible GNS_PDA_ filler on the *R*_c_ of composite films, which can be described as follows (calculated details in Appendix A):(4)$$k-k_{p}=k_{0}{\left[\frac{f-f_{c}}{1-f_{c}}\right]}^{\tau }$$
(5)$$R_{c}=\frac{1}{k_{0}Lf_{c}{}^{\tau }}$$
where *k*_0_ is a pre-exponential factor ratio determined by the interconnected BN networks; *f*_c_ is the measured and critical volume fraction of the hybrid fillers in the films; and *τ* is the conductivity exponent determined by the aspect ratio of the BN. The *f*_c_, *k*_0_, and *τ* values are 15 vol%, 44, and 0.62 for BC/BN_PDA_/GNS_PDA_-5 and 10 vol%, 28, and 0.56 for BC/BN_PDA_, respectively, and *R*_c_ is obtained by combing the above Foygel model equations. As a result, the *R*_c_ between partially connected “BN-GNS-BN” interfaces is calculated to be 0.9 × 10^5^ K·W^−1^, which is approximately half of that of “BN-BN” interfaces (2.1 × 10^5^ K·W^−1^), which demonstrates more contact probability of filler and efficient phonon transport in BC/BN_PDA_/GNS_PDA_-5 films (Figure 3d).

The heat transfer distributions of composite films in the presence and absent of GNS_PDA_ filler is simulated using the finite element, and finite element models, boundary conditions, and mesh distributions were detailed described in Appendix A. Figure 3e displays the calculated heat flux distributions and thermal conductivities of BC/BN_PDA_ and BC/BN_PDA_/GNS_PDA_-5 composite films. For BC/BN_PDA_ film, the BN fillers exhibit obvious randomly oriented distributions, and the heat flux vector is mainly distributed along the BN pathway. Meanwhile, the noncontact BN fillers contain numerous BN–BN interfaces, and large thermal scattering remains at the edges of h-BN fillers. By contrast, for BC/BN_PDA_/GNS_PDA_-5 film, the higher heat flux vector is mainly distributed along the “BN-GNS-BN” pathway due to the adjacent BN_PDA_ fillers are bridged by flexible GNS_PDA_. Additionally, the relative longer and denser thermally conductive pathways along hybrid fillers are formed, which effectively minimizes the phonon interface scattering. The numerical thermal conductivities of BC/BN_PDA_ and BC/BN_PDA_/GNS_PDA_-5 composite films are separately 21.0 and 34.7 W·m^−1^·K^−1^, which are approximately consistent with the experimental values (21.7 and 34.9 W·m^−1^·K^−1^) at the fixed filler content of 70 wt% (Figure 3a). Furthermore, Figure 3f further compares the thermal conductivities of BC/BN_PDA_/GNS_PDA_-5 composite films in this work with BN-based fillers composites from the literature at the fixed filler content [5,19,30,31,32,33,34,35,36,37], which suggests that the BC/BN_PDA_/GNS_PDA_-5 films prepared using present strategy show satisfactory in-plane thermal conductivity.

Given that GNS can bridge BN to extend the “BN” pathway, the aspect ratio of BN fillers should be carefully considered to further improve the performance of the composites. This is because many studies pointed out that the increment of filler aspect ratio will significantly improve the phonon mean free path and the thermal conductive pathway [10], and our recent research also reveals that the thermal conductivity of composites can increase to 330% when the aspect ratio of BN increases from 5 to 25 [8]. However, the most available methods for the preparation of high aspect ratio BN fillers, including ultrasonic, ball milling, and liquid-phase exfoliation methods, generally involve the problems of complex synthesis, cumbersome process steps, and low yields. The challenges in obtaining scalable high aspect ratio of BN fillers makes it hard to directly apply our developed strategy to the industrialized production of composites. Thus, it would be of great interest to further investigate the effect of ultrahigh aspect ratio BN filler on the mechanical, electrical, and thermal properties in the future works.

### 3.3. Mechanical and Dielectric Properties of Composite Films

In addition to the highly anisotropic thermal conductivity of composite films, the good mechanical and electrical insulation is also of great significance for practical applications. Figure 4a,b reveal the tensile strength and elongation at break of the BC/BN_PDA_ and the BC/BN_PDA_/GNS_PDA_-5 composite films at varied filler contents. It can be seen that the pure BC film shows good mechanical strength (66.1 MPa), which is similar to the results found in the literature [5,38]. Evidently, the BC/BN_PDA_/GNS_PDA_-5 films exhibit higher tensile strength and elongation at break relative to BC/BN_PDA_ films. For example, the tensile strength of the BC/BN_PDA_/GNS_PDA_-5 film is 30.9 MPa at 70 wt% filler content, which increases to 156% relative to BC/BN_PDA_ film (19.8 MPa). At the same time, the elongation at break of BC/BN_PDA_/GNS_PDA_-5 film reaches 4.1%, which increases to 205% compared with BC/BN_PDA_ film (2.0%). This can be analyzed through the microstructures in SEM images. On the basis of the “nacre-like” lamellar structures of BC/BN_PDA_ films, there are few interfacial gaps between BN_PDA_ and BC nanofibers (Figure 2a), which destroy the formation of continuous and complete the BC nanofiber networks. With the introduction of PDA-modified GNS filler, the GNS_PDA_ has a better match with BC matrix in intrinsic flexibility and can be completely covered by the BC nanofibers and BN_PDA_ fillers (Figure 2b), thus forming a more continuous network structure within the film. Therefore, the GNS_PDA_ filler can play a flexible bridge role in connecting isolated and brittle BN fillers, thus effectively transferring the stress and delaying the deformation of the composite film. In addition, we also performed the stress−strain experiments on the BC/BN_PDA_/GNS_PDA_-5 composite films at varied filler contents, as presented in Appendix A. The ultimate tensile strength and elongation at break at different filler contents are basically consistent with the experimental results in Figure 4a,b. For example, at the fixed filler content of 70 wt%, the ultimate tensile strength and elongation at break are 29.7 MPa and 4.0%, respectively, which are basically consistent with results from Figure 4a,b (30.9 MPa and 4.1%). Appendix A shows that the BC/BN_PDA_/GNS_PDA_-5 composite films at the hybrid filler content of 70 wt% can easily lift a 500 g weight, indicating the robust mechanical property [39,40]. Figure 5c presents the tensile recovery result of the BC/BN_PDA_/GNS_PDA_-5 film at the fixed filler content of 70 wt%, and it was found that when the tensile strain of the film gradually increases from 1.5 to 4%, the stress of the film at 1.5% strain is 3.0, 4.3, 6.5, 7.4, and 8.5 MPa, respectively, exhibiting an obvious increasing trend. This is because when the film is stretched, the ultrahigh aspect-ratios of the BN_PDA_, GNS_PDA_, and BC matrix will be further distributed along the in-plane orientation, forming a tighter network structure, thus strengthening the mechanical properties.

The reduction of electron delocalization arising from the ionic characteristic of the B−N bond generally brings about large band gap of the BN filler, which provides the excellent electrical insulating features of composites [41]. Conversely, the conjugated structures of GNS has high electron mobility (2 × 10^5^ cm^2^·V^−1^·s^−1^) [24], which leads to poor electrical insulation properties. It was found, surprisingly, that the electrical insulation properties can be regulated by controlling the mass ratio of GNS_PDA_. Figure 4d,e depicts the dielectric constant (*ε*) and dielectric loss (tan*δ*) of composite films at varied filler content. It can be seen that the *ε* and tan*δ* values of BC/BN_PDA_/GNS_PDA_-5 films are clearly reduced with the increase in hybrid fillers contents, which has a similar trend seen in the literature [5,37]. For example, the *ε* and tan*δ* values of the BC/BN_PDA_/GNS_PDA_-5 composite film with 70 wt% hybrid fillers contents separately reach the extremely low values of 6.3 and 0.03 at the frequency of 1000 Hz, which are lower than that of the film with 10 wt% hybrid fillers (8.1 and 0.05). This is because the BN is the main filler, which exhibits an extremely low values of *ε* and tan*δ* (~4, ~5 × 10^−4^) [42]; that being said, some studies have reported that the addition of GNS causes the enhancement of *ε* due to the interfacial polarization effect and the formation of the micro-capacitance [43]. However, for the hybrid filler of the prepared BC/BN_PDA_/GNS_PDA_-5 film, the content of GNS filler (5 wt%) is much lower than that of the BN filler (95 wt%), and the small amount of GNS is isolated by numerous BN fillers, so the GNS filler fails to form micro-capacitances inner composites. However, when the mass ratio of GNS_PDA_ increases to 10 wt%, the *ε* and tan*δ* values for the BC/BN_PDA_/GNS_PDA_-10 film increase to 7.8 and 0.05, which are apparently increased relative to the BC/BN_PDA_/GNS_PDA_-5 film. This result shows that when the amount of GNS fillers exceeds the critical value, GNS fillers might connect to each other, resulting in the interfacial polarization effect [44]. Therefore, when the content of GNP is about 5 wt%, it is beneficial to ensure that the composite film exhibits excellent dielectric properties. Moreover, as presented in Figure 4f, at the fixed GNS_PDA_ of 5 wt%, the AC conductivity of BC/BN_PDA_/GNS_PDA_-5 film can be dramatically decreased with the increase in the hybrid fillers contents, further showing an obvious enhancement of electrical insulation properties [37]. However, when the mass ratio of GNS_PDA_ further increases to 10 wt %, the AC conductivity of the BC/BN_PDA_/GNS_PDA_-10 film further increases to 6.5 × 10^−10^ S·cm^−1^, which is a significant enhancement compared to the BC/BN_PDA_/GNS_PDA_-5 film (1.8 × 10^−10^ S·cm^−1^). This may be because when the mass ratio of GNS_PDA_ is close to a critical value (5 wt%), the electrical conductivity caused by GNS_PDA_ could be largely cut off by the excellent electrical insulation of h-BN filler networks. As a result, the low dielectric constant, dielectric loss, and electrical conductivity, as well as high thermal conductive properties, can be effectively balanced by simply tuning the mass ratio of the supplementary GNS_PDA_ filler.

### 3.4. Heat Dissipation Application of the BC/BN_PDA_/GNS_PDA_-5 Composite Films

Given the highly thermal conductivity and electrical insulation, as well as excellent mechanical robustness, the BC/BN_PDA_/GNS_PDA_-5 composite films have potential applications in the thermal management of cooling high-power electronic devices. Here, the cooling efficiency of three composite films, including pure the BC, BC/BN_PDA_, and BC/BN_PDA_/GNS_PDA_-5 films, is compared through heat dissipation experiments. As displayed in Figure 5a, the LED chip (10 W) was placed in the middle of films, and the aluminum plate was directly mounted on the films to ensure effective heat diffusion. The IR camera was used to record the hot-spot temperature changes and the temperature distributions images of LED chips. As shown in Figure 5b,c, the LED chip achieves a lower temperature of 74.4 °C within 200 s using the BC/BN_PDA_/GNS_PDA_-5 films, which is, respectively, 22 °C and 13.9 °C lower than when using the BC (96.4 °C) and BC/BN_PDA_ films (88.3 °C). It is worth noting that the operation temperature of LED chips closely relates to their working lifetimes; for example, it is reported that every 10 °C drop in chip temperature can extend its service life by 50% [5]. Therefore, the BC/BN_PDA_/GNS_PDA_-5 film can effectively dissipate the heat and quickly reduce the hot spot temperature, which reveals great potential for use in electronic packaging and electrical thermal management applications.

## 4. Conclusions

In summary, we developed a facile route to fabricate large-scale composite films using the assembly of PDA-functionalized GNS and BN fillers, as well as BC nanofibers. The introduction of the PDA layer greatly improved the interface compatibility between the GNS filler, BN filler, and BC matrix, and the presence of GNS_PDA_-bridging significantly increased the probability of effective contact with BN_PDA_ fillers. The reduction in *R*_I_ and *R*_c_ was conducive to the formation of the well stacked and highly oriented microstructure of the BC/BN_PDA_/GNS_PDA_ composite film, which was beneficial when forming a denser and complete “BN-GNS-BN” heat conduction pathway and tight filler–matrix network. For example, the BC/BN_PDA_/GNS_PDA_ composite film shows a thermal conductivity and tensile strength of 34.9 W·m^−1^·K^−1^ and 30.9 MPa, which separately increased to 161% and 155% relative to the BC/BN_PDA_ film. Although the introduction of the high electric conductivity GNS_PDA_ may affect the electrical insulation of composites, it was found that the low electrically conductive and high thermal conductive properties can be well balanced by regulating the critical value of the mass ratio of GNS_PDA_ fillers. At the optimal GNS_PDA_ mass ratio of 5 wt%, the electrical conductivity caused by GNS_PDA_ could be effectively blocked by the BN_PDA_ filler network, thereby resulting in a low electrical conductivity of 1.8 × 10^−10^ S·cm^−1^. Moreover, the BC/BN_PDA_/GNS_PDA_ composite films effectively transfer heat and diminish the hot spot temperature in LED chip cooling applications. The present concise strategy simultaneously meets the high anisotropic thermal conductivity and low electrical conductivity demands, and it may pave the way to promoting the potential of the industrialization of scalable thermal management devices.

## Data Availability

Data are available upon request from the authors.

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
