# Peer review of "Dopamine-Mediated Graphene Bridging Hexagonal Boron Nitride for Large-Scale Composite Films with Enhanced Thermal Conductivity and Electrical Insulation"

_nanomaterials, 2023, doi:10.3390/nano13071210_

Round 1
Reviewer 1 Report
This paper demonstrates the effectiveness of dopamine-mediated graphene bridging boron nitride for enhancement of thermal conductivity and electrical insulation. It is well organized from the design of the material to the demonstration experiment for a device. This reviewer feels that the paper is almost acceptable as it is. However, several shortcomings are fixed before it is accepted.
1) The key material in this work is a composite of BC, BN_PDA, and GNS_PDA. As shown in Figure 2, its structure is not uniform. How do the authors ensure the uniformity of the electrical, thermal, and mechanical properties? This reviewer guesses that those properties are different from place to place.
2) The tensile strength of the material increased to 30.9MPa, which was higher than that of a material without the GNS_PDA. Explain the mechanism in the text, please.
3) By using the material developed, the mechanical, electrical, and thermal properties became better. Especially, the heat transfer performance sounds nice. To improve those properties more, what should be made in the future? Add it in the text, please.
Reviewer 2 Report
Article" Dopamine-Mediated Graphene Bridging Hexagonal Boron Nitride for Large-Scale Composite Films with Enhanced Thermal Conductivity and Electrical Insulation" is interesting and may be of interest to readers of Nanomaterials. I recommend a minor revision.
- Please emphasize the legitimacy and sense of using biomaterials such as polydopamine and bacterial cellulose combined with graphene nanosheets (GNSPDA) and hexagonal boron nitride (BNPDA) and their subsequent recycling.
- Please provide more detailed characterization of bacterial cellulose (BC) (method of obtaining basic properties) and Dopamine (method of obtaining basic properties).
- In the section Mechanical and Dielectric Properties of Composite Films, refer the obtained results to the literature for similar materials.
- Please complete the conclusions with the selection of a specific composition of the composite with the best properties.
Reviewer 3 Report
Although this work could be interesting to the readers of the journal, there are several writing issues that should be fixed during the revision. My other comments are given below:
1- The conclusion of the paper should be written in past tense, but not in present tense. There are several grammatical errors all over the ms that should be fixed.
2- The mechanical properties discussed in section 3.3 are not verified by theory (e.g., by density functional theory calculations). Therefore, I suspect how reliable are the moduli data provided in MPa? And what is physical significance of the dielectric constant presented in this section? I can only see that the authors have discussed an increase/decrease of the dielectric constant with respect to wt%. What is next?
3- No background references are provided while discussing the Raman spectral features shown in Fig. 1a. This should be done in a revision.
4- Scheme 1 cannot be apparently understood. It should be simplified so that one can understand it from the first glance.
Round 2
Reviewer 3 Report
The work may now be considered for publication since authors have revised their paper based on the comments made of my review.